# Simulation Validation of an 8-Channel Parallel-Transmit Dipole Array on an Infant Phantom: Including RF Losses for Robust Correlation with Experimental Results

**DOI:** 10.3390/s24072254

**Published:** 2024-04-01

**Authors:** Jérémie Daniel Clément, Özlem Ipek

**Affiliations:** 1System Technologies, Siemens Healthineers AG, 91052 Erlangen, Germany; 2School of Biomedical Engineering & Imaging Sciences, King’s College London, London SE1 9NH, UK

**Keywords:** simulation validation, co-simulation, RF losses, dipole array, ultra-high field, RF coils, MRI coils

## Abstract

It is crucial to demonstrate a robust correlation between the simulated and manufactured parallel-transmit (pTx) arrays performances to release the currently-used, very restrictive safety margins. In this study, we describe the qualitative and quantitative validation of a simulation model with respect to experimental results for an 8-channel dipole array at 7T. An approach that includes the radiofrequency losses into the simulation model is presented and compared to simulation models neglecting these losses. Simulated S-matrices and individual B1+-field maps were compared with experimentally measured quantities. With the proposed approach, an average relative difference of ~1.1% was found between simulated and experimental reflection coefficients, ~4.2% for the 1st coupling terms, and ~9.4% for the 2nd coupling terms. A maximum normalized root-mean-square error of 4.8% was achieved between experimental and simulated individual B1+-field maps. The effectiveness of the simulation model to accurately predict the B1+-field patterns was assessed, qualitatively and quantitatively, through a comparison with experimental data. We conclude that, using the proposed model for radiofrequency losses, a robust correlation is achieved between simulated and experimental data using the 8-channel dipole array at 7T.

## 1. Introduction

At ultra-high field (≥7T), multi-channel RF coils were shown to provide significant advantages in comparison with default single-channel RF coils to tackle the inherent B_1_^+^-field inhomogeneity [1,2,3,4]. To ensure safe human use of the RF coils when using RF shimming approaches [5,6], numerical simulations of the electromagnetic fields are needed. The specific-absorption-rate (SAR) levels are then calculated, and safe power limits for the RF coil are established [4]. A dipole coil array was previously introduced for infant MR at 7T [7], and initial results for SAR levels were shown. It is, however, crucial to further refine the coil simulation model by demonstrating an improved correlation between the simulated and manufactured parallel-transmit (pTx) arrays performances to release the currently-used, very restrictive safety margins (~5x more restrictive compared to single-Tx for commercial coils). To date, parallel-transmit RF shimming methods are not extensively used in routine MR scans due to heavy local SAR restrictions on the acquisition parameters. To best represent the real coil setup in simulations, different approaches were previously reported [8,9,10]. The CAD model for the RF coil can be derived from CT images [10] or directly exported from any available CAD software used to draw the coil structure. To represent circuit losses, attenuators can be included at the input ports of the RF coils [9], and the first validation step consists of comparing measured and simulated transmit field maps as well as S-matrices. A second validation step may include MR thermometry or thermal simulations to validate the SAR maps [10,11,12]. However, although high-quality modeling is beneficial, the higher the accuracy, the higher the required computational power and simulation time. Key aspects for validating EM simulations were previously described [13]. However, additional design features may also contribute to the obtention of a robust correlation between simulations and measurements. Robust validation results and methods were reported for a loop coil array at 7T [9] and a dipole coil array at 10.5T [10]. However, for the loop coil array validation at 7T, there was no quantitative comparison of the simulated/measured results. With the dipole array at 10.5T, the impact of different co-simulation parameters was investigated, such as the presence of a cable with variable length that helped to match the S-parameters. However, the individual-channel phase maps are not quantitatively compared.

In a previous work, the pTx dipole coil array design approach for infant MR imaging was introduced and compared to other existing coil designs and a preliminary validation of the simulation model with experimental data was performed [7]. In this work, we aim to further investigate the validation of the coil model, as it is an essential step towards the use of the coil array for the in vivo MR imaging of young infants. We studied three simulation setup configurations, including the configuration used in [7], to investigate the role of RF losses (rather than cable phase length and attenuation) in the correspondence of the simulated and experimental scattering matrix and individual and shimmed B_1_^+^ maps with similar amplitude and phase pTx RF pulse settings for the manufactured pTx dipole array with an in-house manufactured phantom. The most accurate result was then used to derive SAR levels in the infant model [7].

## 2. Materials and Methods

A realistic infant-size phantom (Figure 1a) was designed in house (Solidworks 2021, Dassault Systèmes, Vélizy-Villacoublay, France) and 3D-printed (Deed3D Technology Co, Guangzhou, China) in nylon with solid infill property. In addition, the inner surfaces were post-processed with an epoxy resin to ensure full waterproof properties, and outer surfaces were painted. The phantom was filled with a saline solution at a concentration of 5.8 g/L. The total volume was approximately 4.8L, and the dielectric properties (ε_r_ = 79, σ = 0.95 S/m) were experimentally measured using a dielectric assessment kit (DAK 12, SPEAG, Zürich, Switzerland). The aperture was sealed with a transparent cover, which allows air bubbles to be seen, an O-ring, and eight plastic screws.

The built 8Tx-dipole array consisted of eight center-shortened dipole antennas [7] (length = 230 mm, width = 15 mm) etched on a 1.6 mm-thick FR-4 substrate. They were arranged circularly (diameter = 328 mm) at 45 degrees from each other. Figure 1b shows the tuning/matching circuit layout for the dipoles. It consisted of two series inductors (L_1_ and L_2_), two identical series capacitors C_2_, and one parallel capacitor C_1_ (American Technical Ceramics, Fountain Inn, SC, USA). The circuit board was placed 15 mm above the dipole PCB (Figure 1b) with the two inductors placed in the gap. Dipoles were tuned and matched using a 4-channel vector network analyzer (Keysight Technologies E5080A-ENA, Santa Rosa, CA, USA) and the scattering matrix (S-matrix) was experimentally measured at 297.2 MHz using the infant phantom. To be as close as possible to the setup in the scanner, the S-matrix was experimentally measured including the TR switch (MR CoilTech Ltd., Glasgow, UK) placed between the 8Tx-dipole array and the network analyzer. Reflection (S_i,i_) and coupling (S_i,j_) curves were recorded for a frequency span of 100 MHz (250 to 350 MHz).

MR acquisitions were performed using the infant phantom on a 7T MR scanner (MAGNETOM Terra, Siemens Healthcare, Erlangen, Germany) with 8 × 2 kW RF amplifiers in prototype research configuration. The transmit B_1_^+^-field, defined by Equation (1):(1)$$B_{1}^{+}=\frac{1}{2}{\left(B_{1x}-iB_{1y}\right)}^{*}$$
is directly responsible for the MR signal generation [14]. It is therefore a commonly used quantity in MR to quantify the RF coil performance. It is calculated as the sum of the x and y components of the magnetic field (B-field) created by the RF coil. The B_1_^+^-field distribution was experimentally measured and normalized to 1 kW total output power at the RF amplifier in circularly polarized (CP) mode (Δφ = 45° increments from dipole 1 to 8), C2P mode (Δφ = −45° increments from dipole 1 to 8), and one RF shimmed configuration with the actual-flip angle method [15]. Individual B_1_^+^-field maps (magnitude/phase) were acquired [16], and normalized to 1 kW total output power at RF amplifier. The individual phase maps were computed relative to the shimmed combined RF phase map.

The three following simulation configurations were evaluated using a finite-difference time-domain (FDTD) simulation software (Sim4life 6.1, ZMT, Zurich, Switzerland).

**Configuration 1.** The 8Tx-dipole array was realistically modeled, including the FR-4 substrate (ε_r_ = 4, no electrical conductivity) and an exact design of copper traces to provide an accurate placement of all the lumped elements. The conductive parts were defined as lossy metal (σ = 5.8 × 10^7^ S/m). All RF ports (eight sources and 40 lumped elements) were driven individually by a Gaussian excitation centered at 297.2 MHz with a 200 MHz bandwidth for 300 periods with auto-termination when the convergence reached −50 dB. Computations were carried out on a dedicated GPU (Quadro P4000, Nvidia Corp., Santa Rosa, CA, USA). Convergence was usually achieved within 60 periods in ~85 min per port.

A co-simulation approach (Optenni Ltd., Espoo, Finland) was used to optimize the lumped elements’ values and to tune and match the dipoles at 297.2 MHz and 50 Ohms. To do so, the simulated impedance data were exported to the co-simulation software. Each port was defined either as an inductor, capacitor, or source. To accurately represent the built array, RF losses must be considered; either they arise from coaxial cables or from the lumped elements. Since the capacitors have low losses, a general Q-factor of 1500 at 300 MHz was applied [17]. The two series capacitors (Figure 1b, C_2_) were defined using a sub-circuit, which forces the optimizer to assign them identical values, as it is for the built coil array. To mimic the RF losses along the line from RF power amplifiers to the coil input, a series resistor was added to both inductors, as it is expected to appropriately represent the power attenuation (linear attenuation coefficient) from the non-simulated coaxial cables. Figure 1b shows the typical port configuration for a given dipole. The optimizer was, in addition, constrained to assign lumped elements values within a given realistic range with respect to the built coil array.

**Configuration 2.** The 8Tx-dipole array was modeled and simulated as in configuration 1 with the exact circuit layout (Figure 1c). However, contrary to configuration 1, no resistors were added in co-simulation and the capacitors were considered lossless (Figure 1c).

**Configuration 3.** The 8Tx-dipole array was modeled and simulated without modelling the exact circuit layout, but including the FR4-substrate (Figure 1d). Instead, each dipole included only the source which was, thereafter, defined in co-simulation approach as a tuning/matching circuit with one series inductor and one parallel capacitor (Figure 1d). The lumped elements were defined as lossless.

In the three configurations, the magnet RF shield (diameter = 600 mm, length = 1150 mm) was included and defined as perfect electric conductor (PEC). The coil frame was imported for an accurate placement of the dipoles but was not itself simulated, since it is not expected to disturb the RF signal. The coaxial cables were not included in the simulation model. The experimental reflection coefficients (S_i,i_) and 1st (S_i,j_) and 2nd (S_i,j+1_) neighbor coupling values were used as a target for the optimizer. Furthest coupling values were low and were not expected to have a significant influence on the simulation results. The simulated optimized S-matrix was obtained and the relative difference with experimental quantities was calculated in the three configurations. The average relative difference was calculated for the compared quantities (S_i,i_, S_i,j_, and S_i,j+1_) using the following Equation (2):(2)$$Avg=\frac{1}{N}*\overset{N}{\underset{i=1}{\sum }}\left|\frac{S_{i,i}^{simulated}-S_{i,i}^{measured}}{S_{i,i}^{measured}}\right|$$

In Equation (2), the calculation is shown for the reflection coefficients S_i,i_, with N being the total number of channels. The same calculation was used for the other terms. The simulated S-curves were also computed for configuration 1. Simulated individual B_1_^+^-field maps were computed and normalized, per channel, and scaled by the corresponding averaged experimental B_1_^+^-field value. The individual phase maps were reversed (Equation (3)) to obtain the right correlation between experimental and simulated results. In Equation (3), the per-pixel modified value of the B_1_^+^-field value is shown for a single channel, where ϕ is the phase in radians at the n-position. The same operation was applied to the other channels. The individual B_1_^+^-field phase maps were calculated relative to the RF shimmed total phase map.
(3)$$B_{1,n}^{+,corrected\;}=\left|B_{1,n}^{+,original}\right|*\exp\left(-1i*\;\phi_{n}^{original}\right)$$

The three combined maps in CP, C2P, and RF shimmed configurations were also calculated using the individual B_1_^+^-field maps previously computed and for the same RF phases applied in measurements. The combined maps were, in addition, normalized to the peak experimental B_1_^+^-field value. Difference maps were calculated, and the normalized root-mean-square error (NRMSE) was computed between simulated and experimental B_1_^+^-field maps following Equation (4).
(4)$$NRMSE=100*\frac{1}{\max B_{1}^{measured}}\sqrt{\frac{\sum_{pixel}{{\left(B_{1}^{simulated}-B_{1}^{measured}\right)}_{pixel}}^{2}}{\#pixels}}$$

In Equation (4), #pixels are the total number of pixels while the sum is performed over each pixel. The maximum measured value was determined as the 99th percentile value to avoid the measurement errors for the slice shown in Figure 1a. Following the results, the lumped element values obtained in configurations 1 to 3 were further used to assess the SAR levels when the infant model was simulated in cardiac configuration. The initial results in the previous work were obtained with configuration 2.

## 3. Results

Figure 2 shows the experimental and simulated S-matrices for configuration 1. An average relative difference of ~1.1% was found between the simulated and experimental reflection coefficients (S_i,i_), ~4.2% for the 1st coupling terms (S_i,j_), and ~9.4% for the 2nd coupling terms (S_i,j+1_). The experimental/simulated S-curves presented a similar pattern, although the difference between 1st and 2nd coupling curves was more pronounced with the simulated data (Figure 2c–f). The capacitor values for C2 are given in Appendix A for the built coil and simulated coil in configurations 1 and 2. The resistors values R1 and R2 are given in Appendix A for the simulated coil in configuration 1.

In configuration 1, the simulated individual B_1_^+^-field magnitude maps were well correlated with the experimental. The global scaling coefficients applied to simulated individual B_1_^+^-field maps ranged from 0.53 for channel 2 to 0.82 for channel 1. Beyond that, the regions where the B_1_^+^-field is cancelled were remarkably similar (Figure 3a). The NRMSE values for individual B_1_^+^-field maps ranged from 2% for channel 2 to 4.8% for channel 3 (Figure 3b). The individual phase maps were also very similar in experimental and simulated results for most of the dipoles, with an average difference of 15 degrees. A noticeable difference in phase distribution was observed with dipoles 4 and 5. The simulated B_1_^+^-field maps in CP, C2P, and shimmed modes were comparable to the experimental maps, using the same phases. NRMSE values of 6.2%, 10.3%, and 7.5% were observed between experimental and simulated maps in CP, C2P, and shimmed modes, respectively.

In configurations 2 and 3, the averaged relative difference with experimental S-parameters in terms of coupling values was increased by ~247% (~43% versus 12.4%) compared to configuration 1 (Figure 4a,b). Notably, the absolute relative differences for S_i,j_ values were quite high (from 34% to 53%), denoting the large gap between the simulated and experimentally measured coupling values. This was translated into the individual B_1_^+^-field maps with coupled B_1_^+^-field patterns clearly visible (Figure 4c) and NRMSE values which ranged from 3.9% for channel 2 to 6.4% for channel 8 in configuration 3 (Figure 4d). Similar results were obtained in configuration 2 (see Appendix A). Between configurations 1 and 3, the largest differences in magnitude and phase were observed in regions outside of the main B_1_^+^-field distribution pattern for the individual B_1_^+^-field maps (Figure 4g, black lines). A mean absolute difference of 0.75 $\mu T/\sqrt{\mathrm{kW}}$ was measured across all channels. Table 1 shows the NRMSE values for the combined B_1_^+^-field maps in the three configurations and three RF phase shims (CP, C2P, and Shimmed). The three configurations demonstrate equivalent levels of correlation when the individual B_1_^+^-fields are combined.

In infant simulations with configuration 1, the resulting S-parameters showed lower coupling levels (up to −10.6 dB, Figure 5a) compared to configuration 2 (up to −6.5 dB). Although with configuration 2, simulated and measured transmit field maps showed good correlation, such variations in S-parameters would increase the uncertainties evaluation and decrease the SAR performance of the coil. When using the optimized S-parameters settings found with configuration 1, robust correlation is achieved between measured and simulated phantom data. This translated into a 27% decrease of the maximum 10g-averaged local SAR value for the infant simulations as well as slight changes in the SAR distribution map (Figure 5b). Compared to configuration 3, configuration 1 demonstrated a 40% decrease of the maximum 10g-averaged local SAR (Figure 5c).

## 4. Discussion

In this study, we demonstrated the robust validation of the simulation model for an 8-channel dipole array using MR experiments on an infant-sized phantom. In particular, the RF losses and their impact on the scattering matrix were investigated to obtain a good match between simulated and experimental data (configuration 1). Two alternative configurations were investigated where no losses were included (configuration 2) or using a simplified dipole model (configuration 3).

Including a series resistor into the co-simulation circuit model highly contributed to obtain a robust correlation between the experimental and simulated S-matrices. Input power is damped by the resistors and, consequently, the B_1_^+^-field efficiency of the simulated dipoles is decreased. This, in turn, decreases the coupling between dipoles in the simulated model.

Nevertheless, in co-simulation, the experimental S-matrix is the target, rather than the individual B_1_^+^-field efficiency. Thus, while it is possible to reproduce the experimental reflection and coupling coefficients, the B_1_^+^-field efficiency of individual dipoles is not controlled and will depend on the optimal resistor values found by the algorithm. Individual scaling was therefore still required, since not all dipoles are affected similarly by the presence of resistors and their values. The need for scaling also indicates that, while they do not interfere with the S-parameters optimized in this study, additional sources of RF losses may have to be considered. To address this, furthest couplings (S_i,j+2_) could be considered. However, adding more constraints would impair the efficiency of the cost-function to match the 1st and 2nd coupling terms with measurements. In addition, in this work, the phase of the S-matrix was not considered as the software optimizer did not have such a feature. Adding an optimization of the coupling phases may improve the correlation level. In [10], a robust quantitative and qualitative validation was shown for a dipole array at 10.5T including phase optimization of the S-matrix. However, although the simulated S-matrix phases are aligned with measured values, a step was required to derive the complex voltage at the level feed-port of the simulated dipoles. Thereafter, the combined simulated and measured B_1_^+^-field maps are compared. With the approach shown in this study, the measured excitation vector is directly applied to the simulated results. This is an important aspect as it allows a direct comparison of what is measured and what is simulated at the complex value level (rather than only magnitude), providing a strong validation of the simulation model. Nevertheless, the derivation of reliable phase maps from the low flip angle GRE images [16] is only valid when a transceiver is used and when B_1_^−^ and B_1_^+^ field profiles are close. When independent receivers are used, it is not possible to reliably calculate the per-voxel phase of each transmit channel.

The phase correction applied to the simulated B_1_^+^-field maps (Equation (1)) was required, since, by convention, a negative phase shift in the scanner corresponds to a delayed signal while in the simulation software, a delayed signal is defined with positive phase shift. This needs to be performed only once per simulation setup, and the updated quantities can thereafter be used to perform, for example, RF shimming.

The good correlation between individual simulated and experimental phase maps on phantom is an essential step for further use of the simulation model, notably for human simulations. Indeed, to obtain an accurate estimation of the specific-absorption rate, the same phases must be used to obtain the experimental and simulated RF shimmed B_1_^+^-field maps. In configuration 3, the coupling between dipoles was more pronounced in the individual magnitude B_1_^+^-field magnitude maps compared to the experimental data. This did not translate into degradation of the correlation level (in terms of NRMSE) for combined B_1_^+^-field maps for both configurations 2 and 3 compared to configuration 1. It may indicate that sufficient robustness can be achieved when the simulated coil model reproduces the measured individual phase maps with sufficient accuracy. Nevertheless, in configuration 1, the correlation with the experimental data is more clearly achieved in terms of individual complex field patterns (magnitude and phase). This robust result, in turn, benefited the real-case, in vivo use of the RF dipole coil array, as, for example, in the infant simulations which depicted increased plausibility in terms of S-parameters and better confidence in obtaining reliable SAR maps.

## 5. Conclusions

In this study, we introduced a model to include RF losses in simulations for an 8Tx-dipole array. The effectiveness of the model to accurately predict the B_1_^+^-field patterns was assessed, qualitatively and quantitatively, through a comparison with experimental data and with previous results obtained with the same coil array design. The B_1_^+^-field maps, obtained when RF losses are absent, do not accurately represent the coupling between neighboring dipoles. We conclude that, using the co-simulation approach and the proposed model for RF losses, a robust correlation is achieved between simulated and experimental data, which represents a further step towards using the 8Tx-dipole array for young infant MR imaging at 7T.

## Figures and Tables

**Figure 1 sensors-24-02254-f001:**
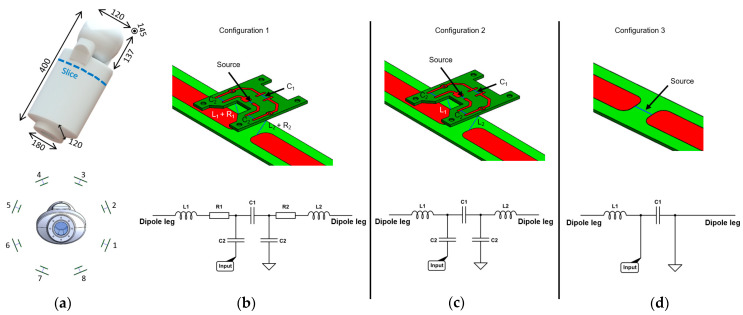
(**a**) 3D CAD model of the infant phantom with general dimensions (in mm) and geometrical arrangement of the dipoles around the infant phantom. Detailed view of the simulated dipole circuit model in configurations 1 (**b**), 2 (**c**), and 3 (**d**) with corresponding lumped elements. In configuration 1, two resistors were added in series with L1 and L2 to mimic RF losses. In configuration 2 (used in [7]), the system was considered lossless. In configuration 3, only co-simulation was used to tune/match the dipoles. Corresponding schematic representation of the tuning/matching circuit are shown for each configuration.

**Figure 2 sensors-24-02254-f002:**
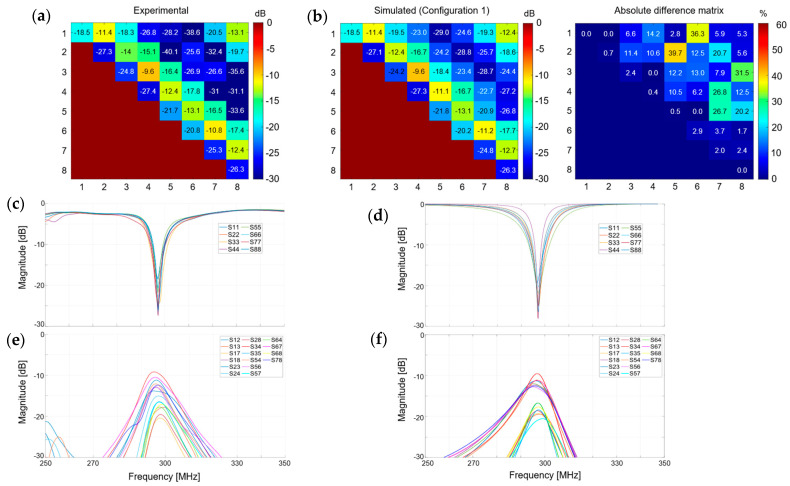
(**a**) Experimental and (**b**) simulated S-matrices and absolute difference matrix obtained for the in-house manufactured phantom (Figure 1a) in configuration 1. An average relative difference of 1.1% was found between simulated and experimental S_i,i_ values, 4.2% for S_i,j_, and 9.4% for S_i,j+1_. (**c**–**f**) Experimental (**c**,**e**) and simulated (**d**,**f**) scattering curves. In (**c**,**d**), the reflection coefficients are shown across a bandwidth of 100 MHz while in (**e**,**f**) the first and second coupling curves are shown.

**Figure 3 sensors-24-02254-f003:**
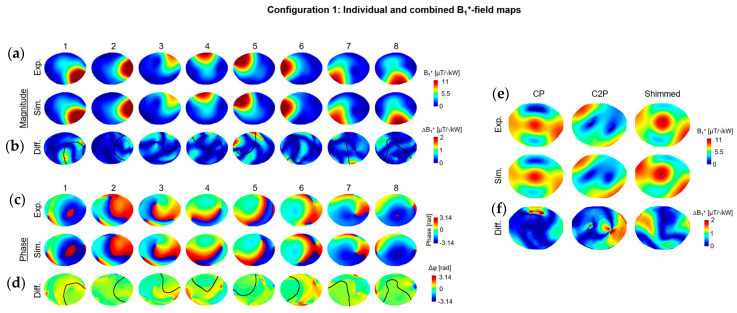
Experimental and simulated B_1_^+^-field maps, normalized to 1kW total input power, shown in (**a**) for individual dipoles and (**e**) for CP and C2P modes and for one RF shimmed configuration. (**c**) Experimental and simulated individual phase maps, computed relative to the shimmed mode. (**b**,**d**,**f**) Difference maps calculated for the combined maps and individual B_1_^+^-field maps with different scale than (**a**,**c**,**e**). In (**b**,**d**) a contour line was drawn (in black) to represent the experimental individual B_1_^+^-field distribution patterns.

**Figure 4 sensors-24-02254-f004:**
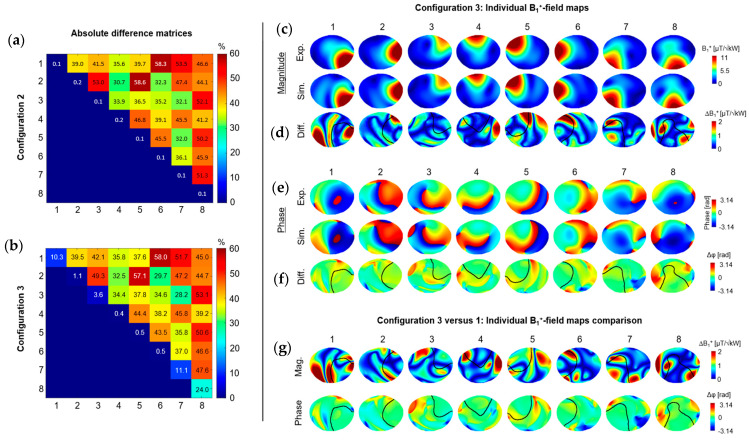
Absolute difference matrix between simulated and experimental S-matrices shown for (**a**) configuration 2 and (**b**) configuration 3. Experimental and simulated magnitude (**c**) and phase (**e**) for individual B_1_^+^-field maps, respectively. (**d**,**f**) Difference maps calculated for the individual B_1_^+^-field maps. In (**d**,**f**) a contour line was drawn (in black) to represent the experimental individual B_1_^+^-field distribution patterns (shown in (**c**)). Color bar for the magnitude differences (**d**) have been rescaled to show residuals more clearly. (**g**) Difference maps between simulated individual B_1_^+^-field maps (magnitude/phase) for configuration 3 and 1.

**Figure 5 sensors-24-02254-f005:**
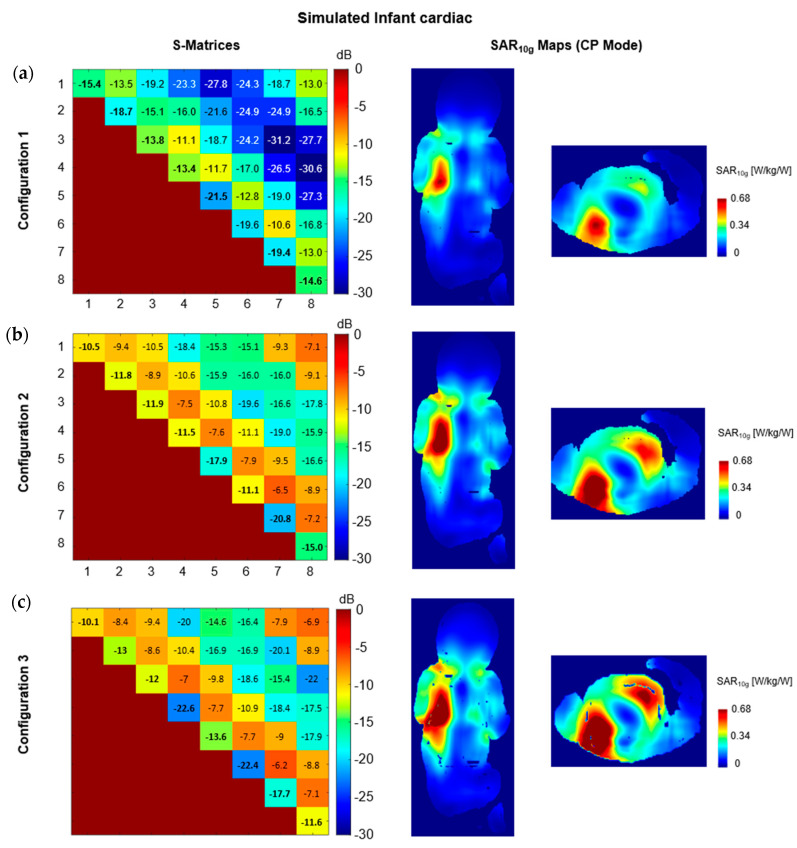
Simulated S-matrix for the infant model, in cardiac configuration, and simulated 10g-averaged SAR map for (**a**) configuration 1, (**b**) configuration 2, and (**c**) configuration 3. Simulated 10g-averaged SAR maps are shown at the maximum position for each configuration and scaled to the maximum 10g-averaged SAR value achieved in configuration 1.

**Table 1 sensors-24-02254-t001:** NRMSE values for the three configurations for the combined field maps. CP, C2P, and shimmed cases are shown.

NMRSE (%)	Configuration 1	Configuration 2	Configuration 3
CP	6.2	7.1	7.6
C2P	10.3	12	10
Shimmed	7.5	6.3	7.4

## Data Availability

Data are contained within the article and Appendix A.

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
