# Peer review of "Simulation Validation of an 8-Channel Parallel-Transmit Dipole Array on an Infant Phantom: Including RF Losses for Robust Correlation with Experimental Results"

_sensors, 2024, doi:10.3390/s24072254_

Round 1

Reviewer 1 Report

Comments and Suggestions for Authors

1. It is an important study as a robust design paper for the 8-channel dipole array. In the 8-channel dipole array design, not only the loss result, but also the reflection loss, gain, and beam width characteristics of each element are important. And the loss, gain, and beam width characteristics when operating together in the eight arrays are also important. Then it is necessary to compare and display its simulation and manufacturing these results together.  

2-1. Why did you adopt the 8x8 array structure in your application?

2-2. Please define the concept before mentioning the unfamiliar B1+-field.

3-1. The difference for Figure 2 (b) is mainly shown in 1-2 (y-axis) and 5-6 (x-axis), what difference is the result, and what does the result mean?

3-2. Why did the results of 1x1 and 8x8 differ from the results of (a) and (b) in Figure 2?

3-3. Need to explain the difference in the results of the lower frequencies of (e), (f) in Figure 2

4-1. Figure 3(b) shows a large difference at No. 5. Why?

4-2. Why did you use uT values in Figure 3 (a, b, e, f)?

4-3. Why are the CP and C2P results different even when the only phase values are different like as 45deg and -45deg in Figure 3 (e)?

4-4. In Figure 3 (f), why did cp perform best?

5. Edit text displayed in the Figure 5 (a) (b).

6. Why was it not considered to analyze in consideration of the phase in the design? (line 276)

Author Response

Please see attachement.

Reviewer 2 Report

Comments and Suggestions for Authors

This is a solid study by adding series resistors into the dipole feeds in electromagnetic simulation to show high similarity between the measured and simulated dipole S-parameters and B1+ fields, which is important for calculating NRMSE between the simulated and experimental B1+. This calculation is crucial for setting the safety limit of excitation power to ensure it does not exceed the allowed 10-gram SAR limits. Here are several suggestions:

1.       Please indicate the optimized and real values of the in series R1, C2 and C2 in configurations 1-3.

2.       The NRMSE is an important value used in the safety validation to calculate the safety factor. Can you create a table to show the NRMSEs in CP, C2P and shimmed mode between the simulated and experimentally measured B1+ fields for configurations 1-3.

3.       Please add Individual B1+ fields magnitude maps (Exp., Sim. and Diff) and phase maps (Exp., Sim. and Diff) for configuration 2 in either figure 4 or in a supplemental figure.

Reviewer 3 Report

Comments and Suggestions for Authors

In this manuscript, Authors propose a model for RF coils which includes losses into account. They showed that simulations including proposed model differ less from the experimental results than lossless models.

The manuscript is well written and supported with good simulation and experimental results. I would have just a few minor comments/suggestions.

1.      All work has been performed on an infant phantom, maybe that should be said in a title of the manuscript

2.      In Figure 1, please, show the arrangement of dipole antennas around phantom

3.      Please, include SAR simulations for all 3 cases (lossless and lossy) and show the difference in SAR predictions so to demonstrate necessity of modeling losses.     

Round 2

Reviewer 1 Report

Comments and Suggestions for Authors

Please add an explanation of what this paper is different from the ref 7 paper published in 2022, and why it is meaningful. Please update the introduction and conclusion based on this.
